# Can non-pharmacological interventions reduce hospital admissions in people with dementia? A systematic review

Richard Packer[1], Yoav Ben Shlomo[2,3], Penny Whiting[2,3]*

1 Department of Health Sciences, University of Leicester, Leicester, United Kingdom, 2 The National Institute for Health Research Collaboration for Leadership in Applied Health Research and Care West (NIHR CLAHRC West) at University Hospitals Bristol NHS Foundation Trust, Bristol, United Kingdom, 3 Population Health Sciences, Bristol Medical School, University of Bristol, Bristol, United Kingdom

* penny.whiting@bristol.ac.uk

## Abstract

### Background

People with dementia who are admitted to hospital have worse outcomes than those without dementia. Identifying interventions that could reduce the risk of hospitalisation in people with dementia has the potential to positively impact on lives of people with dementia. This review aimed to investigate whether there are non-pharmacological interventions that successfully reduce hospitalisation risk, length of stay and mortality in people with dementia.

### Methods

7 electronic databases and trial registries were searched from inception to October 2018. We included randomised controlled trials that evaluated non-pharmacological interventions in out of hospital settings and targeted people with any type of dementia. All stages of the review process were performed by two reviewers. Risk of bias was assessed using the Cochrane Risk of Bias tool. We grouped studies based on intervention: care management, counselling/self-help, enhanced GP services or memory clinics, and physiotherapy or occupational therapy. Data were pooled within intervention categories using random effects meta-analysis.

### Results

There was no evidence that any of the intervention categories were associated with reduced hospital admission or mortality. There was very weak evidence to suggest that care management interventions (mean difference, MD, -0.16, 95% CI -0.32, 0.01), physiotherapy/occupational therapy (MD -0.16, 95% CI -0.36, 0.03) and enhanced GP/memory clinics (MD -0.14, 95% CI -0.31, 0.03) were associated with small reductions in hospital stay. There was no evidence for an effect of counselling/self-help interventions on length of hospital stay.

**Data Availability Statement:** All relevant data are within the paper and its Supporting Information files.

**Funding:** This research was funded by the National Institute for Health Research (NIHR) Collaboration for Leadership in Applied Health Research and Care West (NIHR CLAHRC West) to PW, https://clahrc-west.nihr.ac.uk/. The funders had no role in study design, data collection and analysis, decision to publish, or preparation of the manuscript.

**Competing interests:** The authors have declared that no competing interests exist.

**Abbreviations:** GP, General practitioner; MD, Mean difference; RCT, randomised controlled trials; RR, Relative risk; USA, United States of America; WHO, World health organisation.

## Conclusions

Current evidence from randomised trials suggests no clear benefit or harm associated with any of interventions on risks of hospitalisation, duration of hospitalisation or death. Further research with the primary aim to reduce hospitalisation in people with dementia is required.

## Introduction

Global dementia cases are estimated at 46.8 million people worldwide, forecast to increase to 130 million by 2050. Dementia leads to a loss of both cognitive and physical function without causing rapid mortality [1] necessitating significant levels of care for extended periods of time. [1] This care incurs cost, which, combined with the rising prevalence led the World Health Organization to announce dementia as a public health priority [2].

Whilst rarely being a direct cause of hospital admission, dementia is a significant co-morbidity, increasing the likelihood of attending hospital. This in itself increases morbidity, particularly accelerated cognitive decline, and mortality[3–5]. People with dementia also have a longer length of stay in comparison to those of the same age without dementia [3, 5, 6]. Furthermore, hospitalisation may precede institutionalisation (movement from a personal home to a care/nursing home) [4, 5, 7], which is further associated with worsening cognitive and physical performance and increased mortality [8]. This culminates in both higher in-hospital and higher post-discharge costs for people with dementia compared to others with the same condition. If unnecessary hospitalisation can be reduced appropriately [9], many of the harmful sequelae can be also be reduced, and quality of life of people with dementia will be improved.

Current pharmacological treatment is limited and is focussed around two classes of medication, acetylcholinesterase inhibitors such as donepezil and galantamine and N-Methyl-D-asartic acid (NDMA) receptor antagonists such as memantine [10]. These medications have been shown to be effective at slowing symptom progression but have negligible effect on survival and are not effective for all people with dementia [11, 12]. This makes the use of non-pharmacological interventions particularly appealing given the current limited benefits of pharmacological therapy. Non-pharmacological interventions can be thought of as anything not directly involving a medication; examples include care/case management models (healthcare workers acts as a "care/case manager" attempting to optimise a complex patient's healthcare needs) [13], and enhancing existing community services. In the context of dementia care, non-pharmacological interventions reviews have focussed on moderating problem behaviours, improving cognition or improving general quality of life and activity levels [14]. This review aims to investigate whether there are non-pharmacological interventions that successfully reduce hospitalisation risk in people with dementia.

## Methods

The review followed methods recommended by Cochrane and the Centre for Reviews and Dissemination, [15] and is reported according to PRISMA guidelines [16] (see S1 File for PRISMA checklist), no ethics approval was required for this review.

### Search strategy

We searched the following electronic databases from inception to October 2018: Medline, EMBASE, PsycINFO, PsycEXTRA, CINAHL Plus, Web of Science, and Cochrane Central. We

also searched trial registries—clinicaltrials.gov and the International Clinical Trials Registry Platform. No date or language restrictions were applied. Search strategies were developed for each database and included terms related to dementia, hospitalisation and study type (RCTs and systematic review/meta-analysis); an example of the full search strategy is provided in Appendix A in S1 Appendix. Reference lists of included studies and relevant reviews were screened to identify additional relevant papers.

## Eligibility criteria

We included randomised controlled trials (RCTs) that evaluated non-pharmacological interventions in out of hospital settings and targeted people with any dementia diagnosis. We included studies where interventions targeted staff (e.g. nurses, GPs) and/or carers alongside people with dementia but those with interventions targeting staff or caregivers alone were excluded. Studies that included only people with mild cognitive impairment or conducted in palliative populations were excluded. Studies where people with dementia formed part of a mixed population were included if data could be extracted separately for people with dementia. Data on hospitalisation had to be available as an outcome. We contacted authors for hospitalisation data if reports indicated that these may have been collected.

## Study selection

Titles and abstracts retrieved by the searches were screened independently by two reviewers. Any references considered potentially relevant were obtained as full text articles. These were assessed independently by two reviewers and reasons for exclusion recorded. Disagreements were resolved through discussion or referral to a third reviewer where necessary.

## Data extraction

Data were extracted by one reviewer and checked by another for accuracy using a standardised form. Any disagreements were resolved through discussion with a third person. When a paper quoted a protocol or trial registry number, that paper or trial record was retrieved and used in addition to the included paper to gather all the data possible. In cases where the primary outcome data required clarification or important data were missing, the authors were contacted.

We extracted data on: country of study, study design, study setting, sample size, participant characteristics (age, gender, type of dementia, severity of dementia, existence of comorbidities), details of the intervention and, outcomes assessed. Data to calculate two measures of hospitalisation were extracted, dichotomous data for relative risk (RR) of hospitalisation (defined as one or more hospitalisation in the measurement period) and continuous data (mean difference and standard deviation) for difference in length of hospital stay (MD); where available emergency hospital admission was selected in preference to all admission modalities. Data to calculate a RR for mortality were also extracted (extracted data can be seen in S2 File).

## Risk of bias of individual studies

The Cochrane Risk of Bias Tool was used to assess included studies for risk of bias [17]. The tool includes five domains of bias: selection bias (random sequence generation and allocation concealment), performance bias (blinding of participants and personnel), attrition bias (incomplete outcome data) and reporting bias (selective reporting). We did not assess any additional sources of bias. In each domain, the risk of bias was rated 'high', 'low' or 'unclear'. Each study was also assigned an overall risk of bias–if studies were rated 'high' on at least one domain then the whole study was considered at 'high' risk of bias; if the study was rated

'unclear' on at least one domain the study was considered at 'unclear' risk of bias; all other studies were considered at 'low' risk of bias.

## Synthesis of results

We grouped studies according to intervention type: care management, counselling/self-help, enhanced GP services or memory clinics, and physiotherapy or occupational therapy. We produced forest plots showing relative risks together with 95% confidence intervals (CI) for hospitalisation and mortality and for mean difference with 95% CI duration of hospitalisation. We estimated summary effect sizes using random effect meta-analysis to allow for differences in the treatment effect across studies.[18] Heterogeneity was investigated visually using forest plots and statistically using the $I^2$ statistic.[19]

## Results

The searches identified 5477 unique records, of which 17 trials [20–36] reported in a further 29 publications [37–65] fulfilled our inclusion criteria (Fig 1). A full list of excluded articles and rationale for exclusion is provided in Appendix D in S1 Appendix.

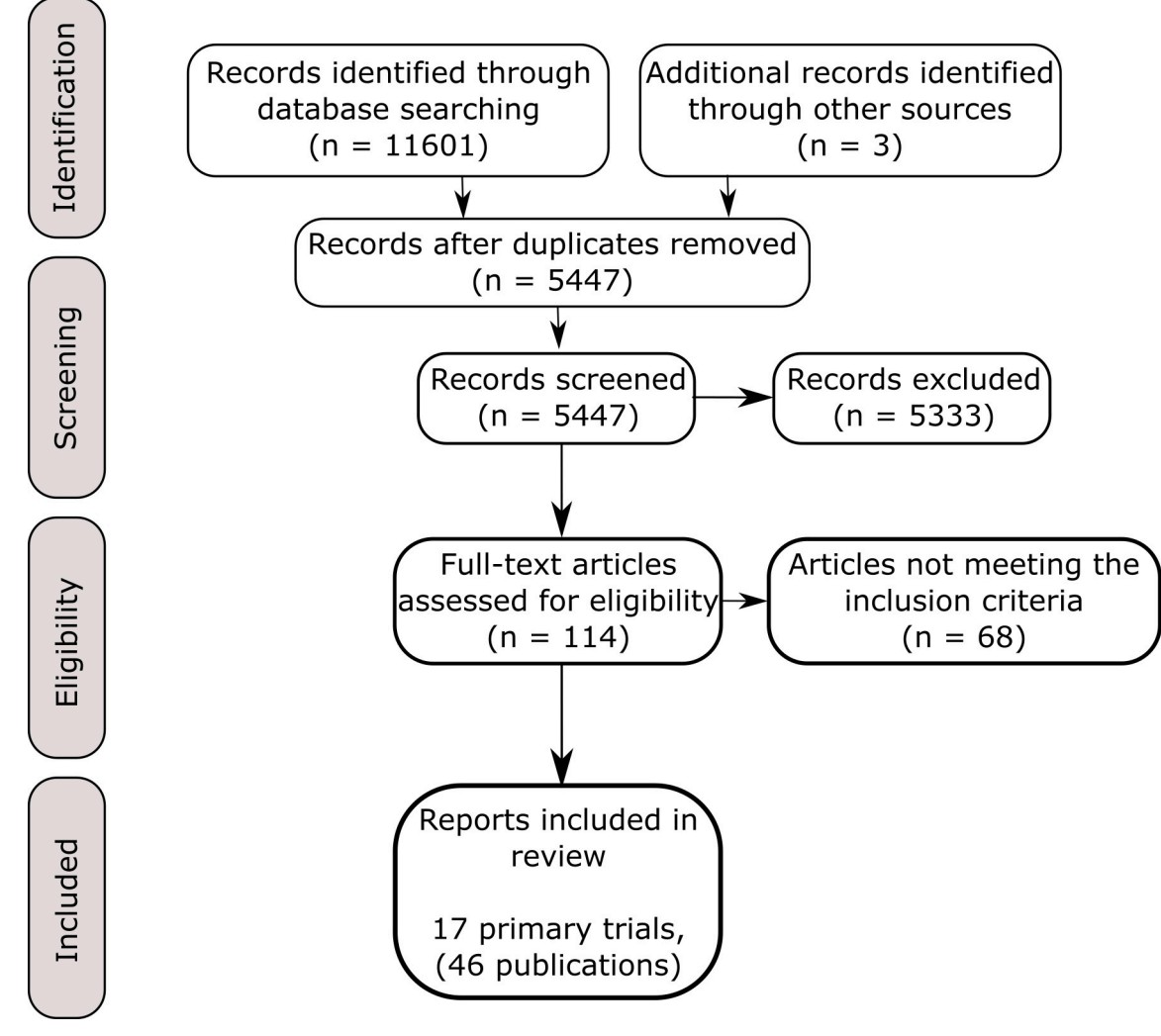

**Fig 1. PRISMA flow diagram.**

Table 1 describes the characteristics of included trials. All were performed in high income countries either in the United States of America (USA) or northern Europe. Five were cluster randomised trials, the most common unit of randomisation was the patient-carer dyad, with many of the studies requiring a carer for inclusion into the study. Fifteen studies provided an average mini mental state score [66]. This showed a wide range of dementia severity, from 13.8 (moderate dementia) to 24 (mild dementia), with the majority being around 18 (moderate dementia). Almost all studies excluded those with a predicted life span of <6 months and those with significant sensorial deficits. Diagnostic criteria varied; ICD 9/10 and the Diagnostic and Statistical Manual of Mental Disorders III and IV were used most frequently. There was a large amount of heterogeneity in the measurement of co-morbidities making the comparison across studies difficult. Type of dementia was recorded in nine (53%) studies. [24–26, 30, 32–36]. Eight of these included two or more types of dementia while one was restricted to participants with Alzheimer's disease. Mean age for dementia patients ranged from 77 to 82

**Table 1. Characteristics of included studies.**

| Broad Type of intervention | Reference and Country | Intervention Description | Age, Sex, and MMSE** Score | Study duration, 1° or 2° Outcome | Unit + N° randomised | Sample size*** |
|---|---|---|---|---|---|---|
| **Care Management** | Amjad 2017, | **Intervention**:18 months of care co-ordination by team (community worker linked to nurse and geriatric psychiatrist). This team identified needs, provided dementia education and skills, coordinating referrals and linkages to other services and care monitoring. | Age: 84 | 18 months | Patient | I:110 |
| | | | % male: 36% | | | C:193 |
| | USA* [20] | | MMSE: 19.1 | Secondary | 303 | |
| | | **Control:** Received the results of Johns Hopkins dementia needs assessment with recommendations for each unmet need and a brief resource guide. | | | | |
| | Callahan 2006, | **Intervention:** Patients received collaborative care management, education on communication skills; caregiver coping skills; legal and financial advice; patient exercise guidelines with a guidebook and videotape; and a caregiver guide provided by the local chapter of the Alzheimer's Association. Care managers were geriatric nurse practitioners. Initial bimonthly meetings then once a month for a year, they identified problems and attempted behavioural solutions and referral for medication. | Age: 78 | 12 months | Physicians | I:84 |
| | | | % male: 57 | | | C:69 |
| | USA* [23] | | MMSE: 18 | Secondary | 153 | |
| | | **Control:** Everyone received 40–90 minutes counselling and advice from a geriatric nurse practitioner and provided written materials and access to the local Alzheimer's chapter, otherwise usual care. | | | | |
| | Duru 2009, | **Intervention:** Patients were assigned a care manager. Care managers performed structure home assessments, identified problems, initiated care plans and sent summaries to home physicians, they provided ongoing care as need and reassessed every 6 months. Additional community services were also made available such as increased respite care. | Age: 80 | Either 12 or 18 months | GP clinics | I:170 |
| | | | % male: 45 | | | C:126 |
| | USA* [24] | | MMSE: NR^x | Secondary | | |
| | | **Control:** Usual care. | | | 408 | |
| | Eloniemi-Sulkava 2009 | **Intervention:** A family care coordinator (a public health nurse advanced 3.5-year training and specific dementia training) created a support plan. Geriatrician provided comprehensive assessments, with goal orientated support group meetings (5/year) for spouse caregivers, with individualized services co-ordinated with the care co-ordinator. | Age: 78 | 24 months | Patient-carer dyad | I: 63 |
| | | | % male: 59 | | | C:63 |
| | | | MMSE: 13.8 | Secondary | | |
| | Finland [25] | | | | 125 | |
| | | **Control:** Usual care. | | | | |
| | Michalowsky 2017, | **Intervention**: Care management by 6 trained dementia nurses, using a computer based interventional management system. Identify unmet needs, task list generated and discussed with multidisciplinary team, treatment plan is generated from this discussion. 6 months 1 visits 1 hour every month from the nurses, following 6 months the task completion was monitored. | Age: 80 | 12 months | GP practice | I:252 |
| | | | % male: 39 | | | C:108 |
| | | | MMSE: 22.8 | Secondary | 136 GP's (634 patients) | |
| | Germany [31] | | | | | |
| | | **Control**: Usual care | | | | |

(Continued)

**Table 1.** (Continued)

| Broad Type of intervention | Reference and Country | Intervention Description | Age, Sex, and MMSE** Score | Study duration, 1° or 2° Outcome | Unit + N° randomised | Sample size*** |
|---|---|---|---|---|---|---|
| **Counselling Self-help** | Bass 2015, | **Intervention:** Partners in Dementia Care provided coaching for patients and caregivers on how to find solutions to daily problems exacerbated by their Alzheimer's. Contacts were, at minimum, once per month over telephone or email. Each patient had two co-ordinators one for medical concerns and one for non-medical concerns | Age: 79 | 12 months | City | I:206 |
| | | | % male: 98% | | | C:122 |
| | USA* [21] | | MMSE: NR[x] | Primary | 508 | |
| | | | | | | |
| | | **Control:** Usual care, both groups received dementia education materials. | | | | |
| | Laakkonen 2016, | **Intervention:** Patients received 8 weekly sessions of a bespoke designed self-help group based on a psychosocial model. Two people trained for 10 days as group facilitators led the sessions. | Age: 77 | 24 months | Patient-carer dyad | I:67 |
| | | | % male: 42 | | | C:69 |
| | | | | | | |
| | Finland [29] | **Control:** Usual care plus a leaflet on nutrition and exercise. | MMSE: 20.8 | Primary | 136 | |
| | Sogaard 2014 | **Intervention:** Individual and group based counselling sessions, plus educational courses and telephone counselling, (The DAISY intervention) over 12 months | Age: 76 | 36 months | Patient-career dyad | I:163 |
| | | | % male: 46 | | | C:167 |
| | | | | | | |
| | Denmark [34] | **Control:** Controls were followed up (3 times), and were interviewed about their current symptoms and daily life and informed about available support programs. Any problems found were referred to health services | MMSE: 24 | Primary | | |
| | | | | | 330 | |
| | | | | | | |
| | Woods 2012, | **Intervention:** Patients and carers participated in joint reminiscence groups (up to 12 dyads) held weekly for 12 weeks with monthly maintenance sessions for a further 7 months. Sessions followed a treatment manual and were led by two trained volunteers, each session lasted 2 hours and rotated weekly topics. | Age: 78 | 10 months | Patient-carer dyad | I:196 |
| | | | % male: 50 | | | C:140 |
| | UK [36] | | MMSE: 19.3 | Primary | 488 | |
| | | | | | | |
| | | **Control:** Usual care. | | | | |
| **Enhanced GP / Memory clinic** | Bellantonio 2008, | **Intervention:** Patients received four systematic, multidisciplinary assessments conducted by a geriatrician or a geriatrics advanced practice nurse, a physical therapist, a dietician and a social worker, during the first 9 months of their residence in assisted living (at 7, 30, 120 and 320 days). | Age: 82 | 9 months | Patient | I:48 |
| | | | % male: 37% | | | C:52 |
| | USA* [22] | | MMSE: 14.8 | Secondary | 100 | |
| | | | | | | |
| | | **Control:** Usual care. | | | | |
| | Kohler 2014, | **Intervention:** The patient was assigned to a full member GP of the Uckermark dementia network. This GP had undergone specialised training in management and diagnosis and was well connected to local specialist. | Age: 78 | 6–12 months | Patient | I:97 |
| | | | % male: 32 | | | C:106 |
| | Germany [28] | | MMSE: 18.9 | Secondary | 235 | |
| | | | | | | |
| | | **Control:** The patient was assigned to associate member GP's analogous to usual care group. | | | | |
| | Meeuwsen 2013, | **Intervention:** Post dementia care performed by a memory clinic, the Memory clinics used the Dutch Institute for Healthcare Improvement guidelines. | Age: 78 | 12 months | Patient-carer dyad | I:83 |
| | | | % male: 47 | | | C:77 |
| | | | MMSE: 22.7 | Secondary | 175 | |
| | Netherlands [30] | **Control:** Post dementia diagnosis care performed by the general practitioner amounting to usual care, GP's used the Dutch general practice and homecare dementia guidelines. | | | | |
| | Schwarzkopf 2011, | **Intervention A:** GP's received additional training in diagnosis (all groups) and treatments (intervention arms only), in addition the intervention arms had rapid access to outpatient dementia specialists and family caregiver support groups. | Age: 80 | 24 months | GP Practice | IA:108 |
| | | | % male: 32 | | | IB:108 |
| | Germany [31] | | MMSE: 18.7 | Primary | 383 | C:167 |
| | | | | | | |
| | | **Intervention B:** had access to a one on one counsellor in addition to that described for intervention A. | | | | |
| | | | | | | |
| | | **Control:** GP's received additional training in diagnosis, otherwise care as usual. | | | | |

*(Continued)*

**Table 1.** (Continued)

| Broad Type of intervention | Reference and Country | Intervention Description | Age, Sex, and MMSE** Score | Study duration, 1° or 2° Outcome | Unit + N° randomised | Sample size*** |
|---|---|---|---|---|---|---|
| **Physio / Occupational Therapy** | Engedal 1989, Norway [26] | **Intervention:** Patients were offered participation at a day-care centre 3 days a week. The centre was staffed with two nurse aids and one occupational therapist, offering social physical and occupational activities. | Age: 80 | 12 months | Patient | I:38 |
| | | | % male: 31 | | | C:39 |
| | | | MMSE: 18 | Secondary | 77 | |
| | | | | | | |
| | | **Control:** Usual care. | | | | |
| | Graff 2008, Netherland [27] | **Intervention:** Patients received 10 sessions of occupational therapy at home over 5 weeks given by well trained (>80 hours and experienced >240 hours training respectively) occupational therapists specialising in dementia. | Age: 78 | 3 months | Patient | I:67 |
| | | | % male: 44 | | | C:65 |
| | | | MMSE: 19 | Secondary | 78 | |
| | | | | | | |
| | | **Control:** Usual care | | | | |
| | Pitkala 2013, Finland [32] | **Intervention A:** Received home visits from a dementia specialist physiotherapist using "goal orientated tailored therapy", 1 hour twice a week. | Age: 78 | 24 months | Patient-carer dyad | IA:70 |
| | | | % male: 43 | | | IB:70 |
| | | | MMSE: 18 | Secondary | 210 | C:70 |
| | | **Intervention B:** Received group based exercises, with 4-hour visits to day centres twice a week, groups of 10 with two specialised physios'. Effective exercise time of 1 hour/session. | | | | |
| | | | | | | |
| | | **Control:** Usual community care but were given oral and written advice from the nurses regarding exercise and nutrition. | | | | |
| | Voigt-Radloff 2011, Germany [35] | **Intervention:** Patients received 10 occupational therapy sessions of 1-hour duration held over 5 weeks within the home environment. This consisted of an assessment phase shared goal setting and treatment phase to reach the one or two goals set. The carer was involved in learning how to supervise, problem solving and coping strategies. | Age: 78 | 12 months | Patient-carer dyad | I:54 |
| | | | % male: 42 | | | C:50 |
| | | | MMSE: 20.4 | Secondary | 141 | |
| | | | | | | |
| | | **Control:** Patients received 1 hour of occupational therapy, from the same occupational therapists, and received a 10-page leaflet detailing advice about physical activity and signposted to local dementia services. | | | | |

* USA = United States of America

** MMSE = Mini-mental state exam

*** I = Intervention arm; C = Control arm

ˣ NR = Not reported

years across studies and both male and female dementia patients were included. Study duration ranged from three months to two years; most were one-year duration.

Five trials evaluated care management interventions targeting people with dementia. [20, 23–25, 31] These involved introducing the role of care manager, a nurse or community worker whose job was to co-ordinate care of dementia patients, optimising clinic appointments and pathways to improve care. Three trials compared this to usual care, two provided additional assessments and advice at the beginning of the trial to control groups. Four trials evaluated counselling or self-help interventions, providing support/self-help groups, coaching and reminiscence therapy[21, 29, 34, 36]. Control groups consisted of: usual care; usual care with additional information on dementia education or nutrition and exercise; and usual care combined

with interviews investigating health needs with referrals based on the needs identified. Four trials evaluated referral to specialised services–either services where GPs had received enhanced training or to memory clinics. [22, 28, 30, 33]. Control groups received either usual care or mildly augmented usual care with the GP's reminded of existing guidelines. Four trials provided additional physiotherapy, occupational therapy or both. [26, 27, 32, 35] Controls received usual care, a lower level of occupational therapy or advice in addition to usual care.

Hospitalisation was the primary outcome in four trials [21, 22, 24, 26]; in other studies it was included as a secondary outcome, often to inform an economic evaluation. Hospitalisation was measured through direct access to electronic records in four trials [25, 29, 32, 33], through surveys or interviews with patients or carers in nine [20, 22–24, 27, 30, 31, 35]v[36] trials, or both in two trials [21, 34]. Two trials did not report how hospitalisation was assessed [26, 28]. Seven of the seventeen included trials provided additional data after correspondence with the authors [20, 25, 29, 31–34], four of which initially presented no data on hospitalisation [20, 31, 33, 34].

## Risk of bias in included studies

Fifteen trials were considered at high risk of bias overall and two at unclear risk of bias; none were at low risk of bias (Fig 2) (full risk of bias assessment can be seen in Appendix B in S1 Appendix). The main limitation in included studies was lack of blinding of participants and personnel– 12 were judged at high risk of bias and the other five at unclear risk of bias for this domain. The interventions were often pragmatic in nature and so it was not feasible to blind participants to treatment allocation. Outcome assessment was blinded in eight trials and a further seven trials did not report sufficient information to judge this domain. Only two trials were at high risk of bias for random sequence generation and allocation concealment. The available study protocols or other additional papers enabled us to rule out selective reporting bias in ten trials, however, for five trials this could not be ruled out and for two trials there was evidence of selective reporting bias.

## Effect of interventions

There was no evidence that any of the intervention categories were associated with a decreased risk of hospital admission with RR ranging from 0.95 to 1.05 (Fig 3). There was also no evidence of reduced mortality–RRs ranged from 0.62 to 1.15 across studies (Fig 4). There was very weak evidence to suggest that care management interventions (summary MD -0.16, 95% CI -0.32, 0.01; 4 trials, $I^2$ 41.3%), physiotherapy/occupational therapy (summary MD -0.16, 95% CI -0.36, 0.03; 3 trials, $I^2$ 0%) and enhanced GP/memory clinics (summary MD -0.14, 95% CI -0.31, 0.03; 1 trial, $I^2$ 0%) were associated with small reductions in hospital stay (Fig 5). There was no evidence for an effect of counselling/self-help interventions on length of hospital stay. One trial of a care management intervention reported reduced hospitalisation in the group that received care management (RR 0.46, 95% CI 0.23 to 0.92) and reduced number of nights spent in hospital (MD -0.34, 95% CI -0.57, -0.12) (Michalowsky et al., 2017). All other trials found no difference between intervention and control groups (Figs 3 and 4).

## Discussion

This review evaluated whether non-pharmacological interventions can reduce hospitalisation in people with dementia. We identified 17 trials evaluating a broad range of interventions including physio/occupational therapy, enhanced GP services/memory clinics, counselling/ self-help and care management. Overall there was no evidence for an effect of any of the intervention categories on hospitalisation or mortality. The consistency of the effect estimates

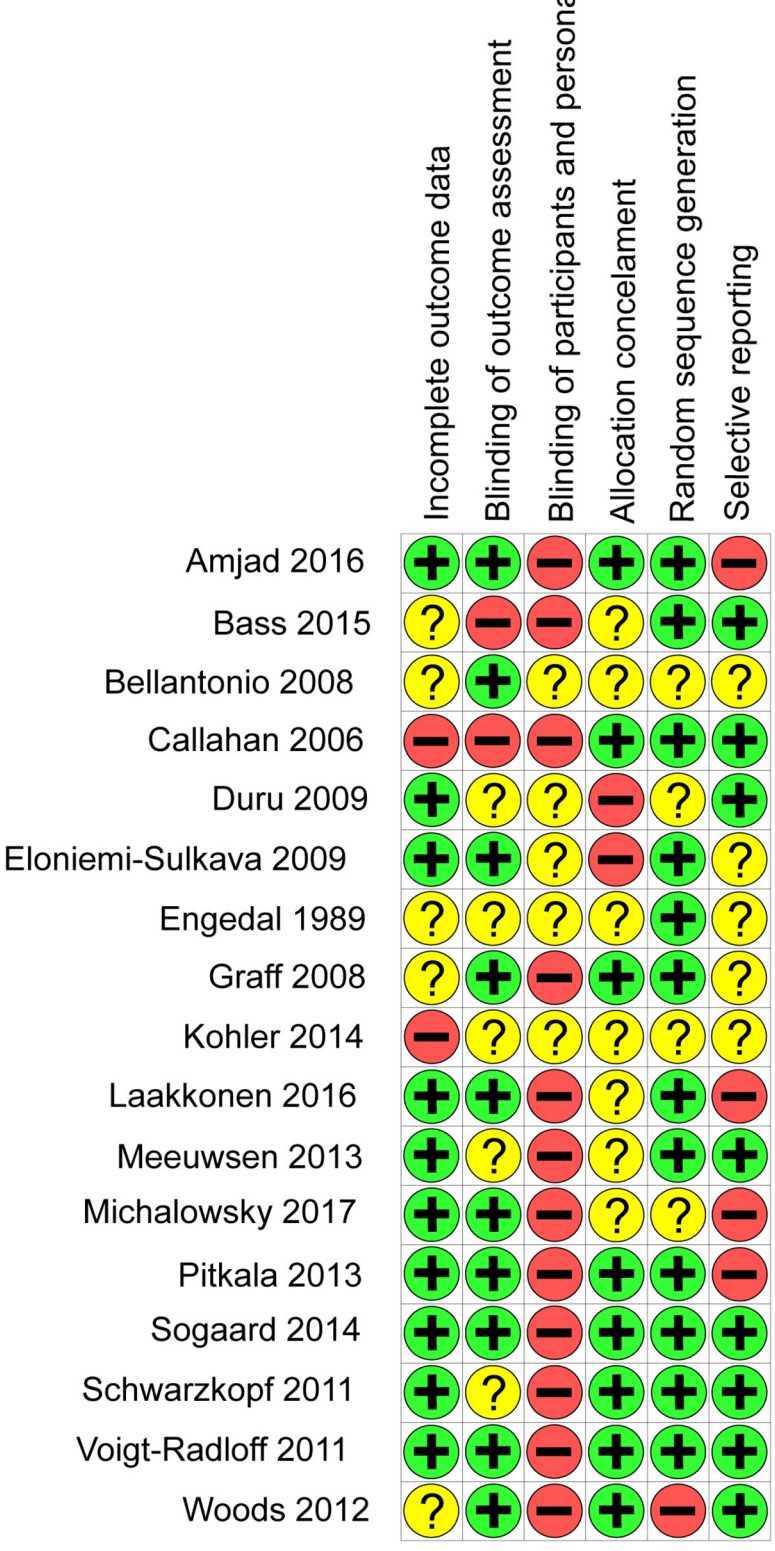

**Fig 2. Result of risk of bias assessment.** + = Low risk of bias,− = High risk of Bias, ? = Unknown risk of bias.

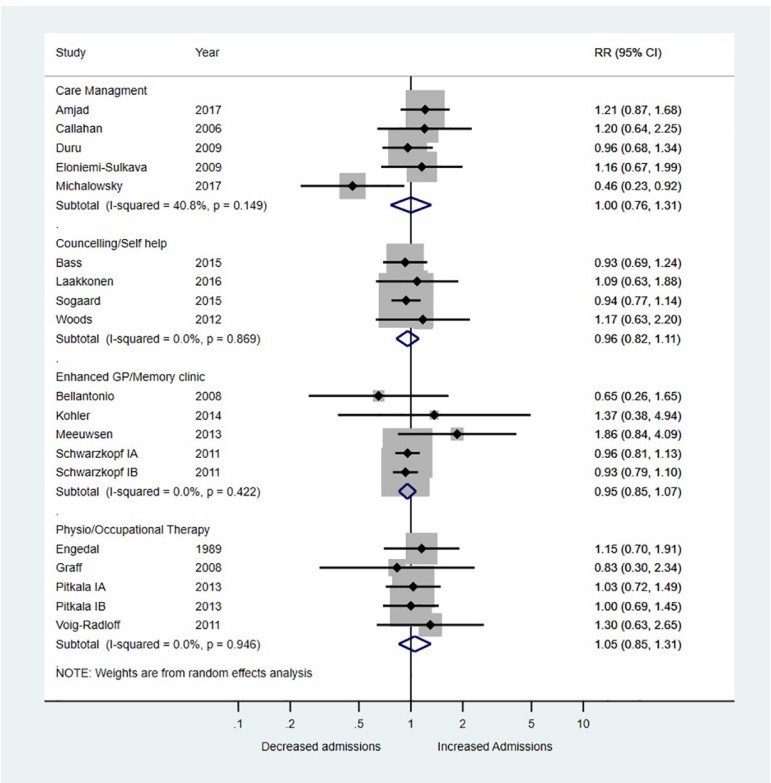

**Fig 3. Forest plot of the relative risk (RR) of hospitalisation stratified according to type of intervention.**

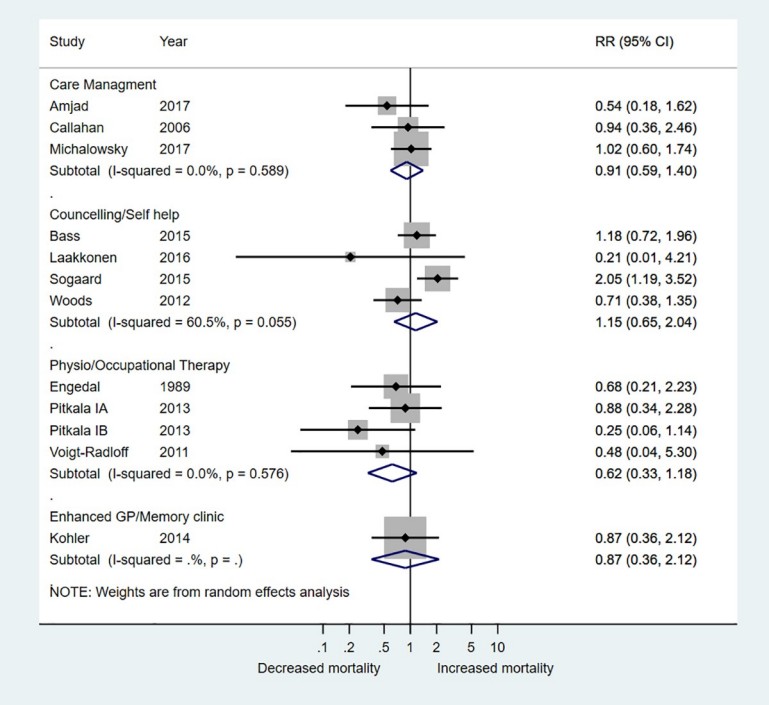

**Fig 4. Forest plot relative risk (RR) of death stratified according to type of intervention.**

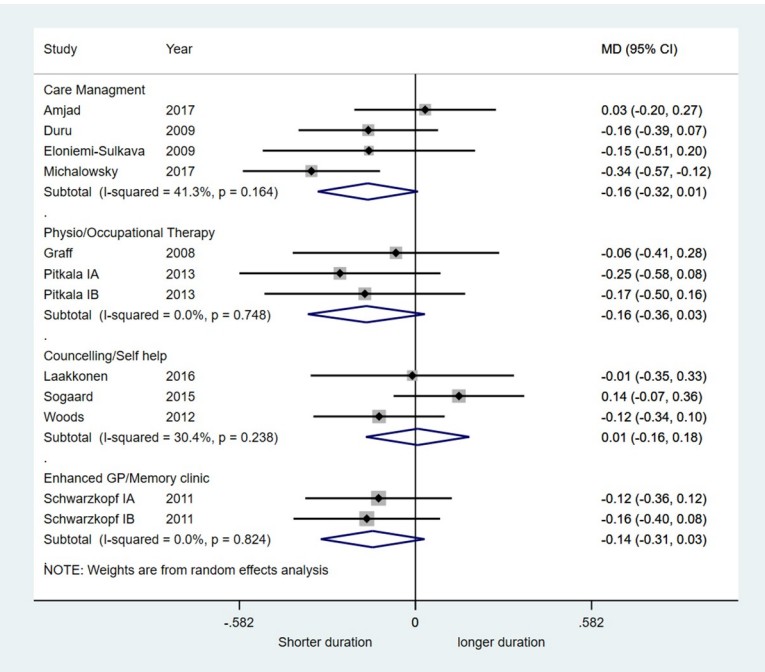

**Fig 5. Forest plot of the mean difference (MD) of nights spent in hospital stratified according to type of intervention.**

suggests that the type of intervention is not important. There was weak evidence for a very small reduction in duration of hospital stay for care management interventions, physio/occupational therapy and enhanced GP services/memory clinics.

Only one trial, which evaluated a care management intervention, showed a beneficial effect of the intervention in reducing the risk of hospital admission and number of nights spent in hospital [31]. The publication of this study did not report on hospitalisation, and only collected these data for cost analyses–we obtained the data on hospitalisation direct from the authors. The trial included patients with mild dementia and it may be that the effect would have been different for more severe cases. Care was managed by trained dementia nurses which may have also contributed to the beneficial outcome seen with the intervention. There were a large number of withdrawals from the study (43% at 12 months) which could have biased the results.

Another trial showed an increased risk of mortality with a regular counselling and self-help intervention that included monthly phone calls to patients and caregivers, and written information and taught courses on the disease [34]. The intervention group in this study were worse off in some respects at baseline such as living alone, renting instead of owning their house and quality of life. These could have contributed to the poorer outcome at the end. Also, the control group was not without intervention and 'structured support' was provided at each follow up session which accommodated the patient's and the caregiver's frustration and uncertainty associated with a recent diagnosis by providing guidance and contact with the relevant local support programmes. This structured support may be something on its own to study in the future for a potentially beneficial effect. These findings from single trials need confirmation with further testing before any conclusions can be drawn.

## Strengths and weaknesses

This review followed guidelines for the conduct and reporting of systematic reviews. [15, 16] We conducted a comprehensive, sensitive search in order to identify as many eligible studies as possible and minimise the risk of publication bias by searching trial registries and contacting authors for unpublished studies and data. Our primary outcome of interest was risk of hospitalisation. We included studies that reported data on hospitalisation, or where authors were able to provide unpublished data. Only four of the included trials reported hospitalisation as a primary outcome, and for the rest it was a secondary outcome with seven studies requiring data from authors. Where hospitalisation was not the primary outcome, this suggests that the intervention may not have been specifically designed to reduce the risk of hospitalisation and so such trials may be less likely to report an effect on this outcome. However, the only trial to find a beneficial effect of the intervention on hospitalisation was one where unpublished data were obtained direct from the authors. [31] Despite our sensitive search, it is possible that we have missed relevant studies where data on hospitalisation were not reported in the publication but were collected as part of the study.

We applied pre-defined, explicit inclusion/exclusion criteria and involved two reviewers in each stage of the review process minimising the potential for bias and errors. We used the Cochrane risk of bias tool [17] to assess included trials. This identified a number of limitations in the included trials, with none judged at low risk of bias across all domains. The main limitation was lack of blinding of participants and personnel. Given the nature of the interventions evaluated, blinding is unlikely to have been feasible in most trials and as hospitalisation and mortality are objective outcomes, the potential for introduction of bias through lack of blinding is less than had more subjective outcomes been measured. We considered the interventions too diverse for an overall summary effect estimate to be appropriate. Instead we grouped studies according to intervention category and used random effects meta-analysis to pool studies within intervention categories, we did not pre-register this analysis.

## Comparison with existing literature

We are aware of one recent review that has assessed a similar topic published in 2015 [9]. Inclusion criteria were similar for both reviews except that the Phelan review did not restrict based on study design but was restricted to studies published in English. We did not apply any language or publication restrictions and used a sensitive search strategy, but we limited inclusion to randomised trials to restrict the review to the highest level of evidence. This is likely why we located 12 times as many citations in our search yet included only six from their set of 10 included studies.

The Phelan review did not formally assess the risk of bias in included studies and did not carry out any statistical synthesis across studies. Very limited information was provided on study results or design. The Phelan review included 10 studies, six of which were also included in our review. The four studies included in the Phelan review not included in our review used a non-randomised design and so did not fulfil our inclusion criteria. We included a further 11 studies not included in the Phelan review. Three studies were published after the Phelan review and so would not have been available at the time that the searches for this review were conducted. A further two did not report data on hospitalisation, we contacted authors to obtain this information whereas reporting this information was a requirement for inclusion in the Phelan review. It is unclear why the other six trials were not included in the review as they appeared to fulfil the review inclusion criteria. However, despite the differences between review methods, overall conclusions were similar with regards to hospitalisation. We also considered duration of hospital stay which was only reported for one study in the Phelan review.

A Cochrane review on case management interventions [67] included some of the studies we included as well, although the overlap is limited because of differences in the set of inclusion criteria between the reviews. They found no effect at any length of follow up (6, 12, and 18–24 months) on hospitalisation risk or mortality but there was a small negative effect of case-management interventions provided in community settings on length of hospital stay (mean difference in number of nights was 0.63 (95% CI 0.40 to 0.86; 3 studies).

## Implications for practice and research

Current evidence from randomised trials suggests no clear benefit or harm associated with any of interventions on risks of hospitalisation, duration of hospitalisation or death. Until better quality evidence becomes available no change in practice based on evidence is warranted.

None of the seventeen trials explicitly targeted the common causes of hospitalisation in people with dementia, either co-morbid conditions such as heart failure or more acute problems such urinary tract infections or pneumonia [68]. Future high quality trials targeting the common causes of hospitalisation in people with dementia, testing new community based interventions are needed. Because hospitalisation is an important outcome for people with dementia this should be explored as a primary outcome of such interventions. It may be that non-pharmacological interventions have no impact on risk of hospitalisation but may affect duration of stay, for example through earlier detection and treatment. This outcome also needs exploring in future studies.

## Conclusions

There is little evidence that any of the included non-pharmacological interventions can reduce the risk of hospitalisation in people with dementia. Care management provides a plausible mechanism for reducing acute admissions to hospital, by proactively managing the care of a person with dementia, they can be reviewed appropriately, and problems identified before they require hospital admission. However, the available evidence is not conclusive on this. Further research with the primary aim to reduce hospitalisation in people with dementia, with explicit attempts at reducing the most common causes of hospitalisation in this group is required.

## Supporting information

**S1 Appendix.** Appendix A-E.
(DOCX)

**S1 File. PRISMA 2009 checklist.**
(DOCX)

**S2 File. Data extracted.**
(XLSX)

**S3 File. Do-file for analysis.**
(DO)

## Acknowledgments

We would like to thank Alison Richards for conducting the literature searches and Sharea Ijaz for assisting in data screening and extraction.

## Author Contributions

**Conceptualization:** Richard Packer, Yoav Ben Shlomo.

**Formal analysis:** Richard Packer, Penny Whiting.

**Methodology:** Penny Whiting.

**Software:** Penny Whiting.

**Supervision:** Yoav Ben Shlomo, Penny Whiting.

**Visualization:** Penny Whiting.

**Writing – original draft:** Richard Packer, Penny Whiting.

**Writing – review & editing:** Richard Packer, Yoav Ben Shlomo, Penny Whiting.

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
