## [Decision Letter · Decision Letter 0]

30 Jul 2019

PONE-D-19-18659

Can non-pharmacological interventions reduce hospital admissions in people with dementia? A systematic review

PLOS ONE

Dear Dr. Richard Packer,

Thank you for submitting your manuscript to PLOS ONE. After careful consideration, we feel that it has merit but does not fully meet PLOS ONE’s publication criteria as it currently stands. Therefore, we invite you to submit a revised version of the manuscript that addresses the points raised during the review process.

ACADEMIC EDITOR: Although it is of interest, the reviewers have raised a number of points which we believe major modifications are necessary to improve the manuscript, taking into account the reviewers' remarks.  

We would appreciate receiving your revised manuscript by Sep 13 2019 11:59PM. To enhance the reproducibility of your results, we recommend that if applicable you deposit your laboratory protocols in protocols.io, where a protocol can be assigned its own identifier (DOI) such that it can be cited independently in the future. For instructions see: http://journals.plos.org/plosone/s/submission-guidelines#loc-laboratory-protocols

We look forward to receiving your revised manuscript.

Kind regards,

Wisit Cheungpasitporn, MD

Academic Editor

PLOS ONE

Journal Requirements:

Reviewers' comments:

Reviewer's Responses to Questions

**Comments to the Author**

1. Is the manuscript technically sound, and do the data support the conclusions?

Reviewer #1: Yes

Reviewer #2: Yes

Reviewer #3: Partly

2. Has the statistical analysis been performed appropriately and rigorously? 

Reviewer #1: Yes

Reviewer #2: Yes

Reviewer #3: Yes

3. Have the authors made all data underlying the findings in their manuscript fully available?

Reviewer #1: Yes

Reviewer #2: Yes

Reviewer #3: Yes

4. Is the manuscript presented in an intelligible fashion and written in standard English?

Reviewer #1: Yes

Reviewer #2: Yes

Reviewer #3: Yes

5. Review Comments to the Author

Reviewer #1: The manuscript is good but not for this journal, Other journals are more appropriate

Reviewer #2: This is a timely systematic review on the use of non-pharmacological approaches to reduce the risk of people with dementia being hospitalized, or having adverse outcomes (length of stay, death) in this setting.

The rationale for the study is well outlined, and the investigators used well established systematic review methods. There are a good number of studies included in the final analysis, and the results are well discussed.

I understand that the dominant approach to systematic reviews is a focus on RCTs, but there is also a substantial literature on the experience of people with dementia in hospitals that speaks to this area. Can I suggest that the authors consider alternate approaches on lessening the impact of having dementia on hospitalization that appears in the literature, perhaps by way of examples in the Discussion?

Is there potential to determine whether the interventions may have had an effect on planned vs unplanned hospitalization?

Reviewer #3: Manuscript is originally written and very well written. It is an interesting and relevant article. I consider it a useful contribution in its field. However, there are many important points that the investigators need to clarify and revise.

1. In abstract, the investigators states “November 2019” which is in the future and not correct. In the full manuscript, the investigators then described “October 2018”.

2. This meta-analysis has not been registered online. Please add this point in the limitation.

3. Figure1, suggest to use PRISMA 2009 Flow Diagram platform

4. Figure 2, please make the colors in the standard per Cochrane; red, yellow, green

5. PRISMA checklist needs to be cited and attached.

6. Who are two independent investigators?

7. It will be better to show kappa for the selection and data extraction. Please show the data of kappa of agreement during the systematic searches. How disagreements were solved during the systematic search among two independent reviewers?

8. Random or Fixed effect was used, needs to be specified in the abstract.

9. Authors should discuss the reason of heterogeneity.

10. Because the protocol has not been registered, please make the data for this review publicly available, possibly through the Open Science Framework (osf.io). Items to include: list of excluded studies, commands for statistical analysis, spreadsheets or data used for the meta-analyses, etc. Making data publicly available will promote the reproducibility of the review and is best practices for systematic reviews and meta-analyses.

6. PLOS authors have the option to publish the peer review history of their article (what does this mean?). If published, this will include your full peer review and any attached files.

Reviewer #1: No

Reviewer #2: No

Reviewer #3: No

---

## [Author Response · Author response to Decision Letter 0]

12 Sep 2019

Response to reviewers

Our response to the reviewers is separated out by reviewer comment, each reviewer’s comments have been labelled R1-3 for reviewer one to three respectively, with each comment a decimal after that, for example R2.1 is reviewer two comment one. The reply to each comment is then below that. This has been done to make reading the reply easier for each reviewer.

R1.1: The manuscript is good but not for this journal, Other journals are more appropriate

We thank reviewer 1 for their comments and their opinion that this is a good paper. They do not explain why they feel that PLOS One is not appropriate but in our opinion, this is an editorial rather than a reviewer decision. The role of reviewers is to aid the editor by commenting on the scientific value of the work (hence whether it should or should not be accepted). The editorial team will screen all manuscripts and will not usually send out papers that they do not feel are appropriate for their journal.

R2.1: This is a timely systematic review on the use of non-pharmacological approaches to reduce the risk of people with dementia being hospitalized, or having adverse outcomes (length of stay, death) in this setting.

R2.2: The rationale for the study is well outlined, and the investigators used well established systematic review methods. There are a good number of studies included in the final analysis, and the results are well discussed.

We thank reviewer 2 for these comments on both the timeliness and methodology of this manuscript, we have taken great effort to ensure the maximum number of studies could be included, including many papers after direct contact with authors to extract unpublished data.

R2.3: I understand that the dominant approach to systematic reviews is a focus on RCTs, but there is also a substantial literature on the experience of people with dementia in hospitals that speaks to this area. Can I suggest that the authors consider alternate approaches on lessening the impact of having dementia on hospitalization that appears in the literature, perhaps by way of examples in the Discussion?

Regarding the inclusion of non-RCT data, as noted by the reviewer this is the predominant for of research including in systematic reviews, we made the decision to include only RCT’s deliberately to ensure any findings we did make were based on the most robust evidence. What this has allowed us to do is make clear statements about the availability of high-quality evidence for the very specific question of evidence of non-pharmacological interventions that reduce the risk of hospitalization. We do briefly discuss potential avenues for further research but are acutely aware that this is not the question that paper is trying to answer and do not want to overstate an opinion not grounded in the evidence we found. We would be happy to be directed towards any pertinent papers that could help inform this part of the paper, although it would be limited to potential avenues for further intervention and not, as could be interpreted by this comment, ways of lessening the impact of dementia once hospitalized.

R2.4: Is there potential to determine whether the interventions may have had an effect on planned vs unplanned hospitalization?

Within some of the studies there was a distinction made between routine outpatient procedures or planned admissions and those that were deemed emergency or unplanned admissions. Within the paper there is a preference to only using figures for emergency admission where this information is available. This has now been made clearer in the methods.

R3.1: Manuscript is originally written and very well written. It is an interesting and relevant article. I consider it a useful contribution in its field. 

We thank reviewer 3 for these comments. 

R3.2: In abstract, the investigators states “November 2019” which is in the future and not correct. In the full manuscript, the investigators then described “October 2018”.

This is now corrected.

R3.3: This meta-analysis has not been registered online. Please add this point in the limitation.

This has now been added see end of strengths and weakness.

R3.4: Figure1, suggest to use PRISMA 2009 Flow Diagram platform

PLOS1 make restrictions on the resolution and format of their figures. To comply with those regulations a figure was created in adobe illustrator that replicated the PRISMA 2009 flow diagram, including the same four headings with virtually identical titles in the boxes. We are happy to submit a word document figure if the editor is willing to accept it.

R3.5: Figure 2, please make the colors in the standard per Cochrane; red, yellow, green

This has been changed, as suggested.

R3.6: PRISMA checklist needs to be cited and attached.

The PRISMA checklist is included, and can be found in S1 Appendix A. It is referenced within that appendix. It is cited in text, however on review that citation is incorrect and has now been corrected.

R3.7: Who are two independent investigators?

The two independent investigators for screening and extraction were Richard Packer (1st author) and Sharea Ijaz (see Acknowledgments)

R3.8: It will be better to show kappa for the selection and data extraction. Please show the data of kappa of agreement during the systematic searches. 

Many thanks for this suggestion, unfortunately it is now no longer possible to go through previous databases and calculate the exact Kappa value for agreement as suggested. There was a robust process including the use of two independent reviewers with a third to act as arbiter when consensus was not present. 

R3.9: How disagreements were solved during the systematic search among two independent reviewers?

Disagreements were resolved via a third person, Penny Whiting (final author). This process is described in the data extraction section of the method.

R3.10: Random or Fixed effect was used, needs to be specified in the abstract.

The analysis was using a random effects, this is stated in the last line of the methods section of the abstract. 

R3.11: Authors should discuss the reason of heterogeneity.

Many thanks for this comment, the I2 statistic is now reported within the text for the highlighted results. Within the discussion we do discuss reasons why the only article to show a statistically significant difference in hospitalization may be different from the other included trials. It formed part of the care-management grouping which was the only grouping to show an elevated I2, in effect describing causes for possible heterogeneity directly without using the statistic.

R3.12: Because the protocol has not been registered, please make the data for this review publicly available, possibly through the Open Science Framework (osf.io). Items to include: list of excluded studies, commands for statistical analysis, spreadsheets or data used for the meta-analyses, etc. Making data publicly available will promote the reproducibility of the review and is best practices for systematic reviews and meta-analyses.

This is an excellent suggestion. We have made the data used for our meta-analysis available within this submission, see S3 Data Extracted. We have also included a list of studies excluded at full text stage see S2 Appendix D. The only thing not currently included are the STATA terms used to create the forest plots. These can be included within the current appendix structure and have been added as S4 Do files for analysis. Given that all the data is available within the PLOS 1 submission framework, we would suggest that there is not a need to repeat this process on the Open Science Framework.

---

## [Decision Letter · Decision Letter 1]

27 Sep 2019

Can non-pharmacological interventions reduce hospital admissions in people with dementia? A systematic review

PONE-D-19-18659R1

Dear Dr. Richard Packer,

We are pleased to inform you that your manuscript has been judged scientifically suitable for publication and will be formally accepted for publication once it complies with all outstanding technical requirements.

With kind regards,

Wisit Cheungpasitporn, MD, FACP

University of Mississippi Medical Center

Twitter: @wisit661 Email: wcheungpasitporn@gmail.com 

Academic Editor

PLOS ONE

Additional Editor Comments:

I want to commend the authors on their superb efforts to revise the manuscript according to all reviewers’ suggestions. The quality of the manuscript has improved substantially.

Reviewers' comments:

Reviewer's Responses to Questions

**Comments to the Author**

1. If the authors have adequately addressed your comments raised in a previous round of review and you feel that this manuscript is now acceptable for publication, you may indicate that here to bypass the “Comments to the Author” section, enter your conflict of interest statement in the “Confidential to Editor” section, and submit your "Accept" recommendation.

Reviewer #2: (No Response)

Reviewer #3: All comments have been addressed

2. Is the manuscript technically sound, and do the data support the conclusions?

Reviewer #2: Yes

Reviewer #3: Yes

3. Has the statistical analysis been performed appropriately and rigorously? 

Reviewer #2: Yes

Reviewer #3: Yes

4. Have the authors made all data underlying the findings in their manuscript fully available?

Reviewer #2: Yes

Reviewer #3: Yes

5. Is the manuscript presented in an intelligible fashion and written in standard English?

Reviewer #2: Yes

Reviewer #3: Yes

6. Review Comments to the Author

Reviewer #2: (No Response)

Reviewer #3: I have no competing interests. All my concerns have been fully elucidated, missing sections and analyses have been completed. Finally, comprehension errors have been corrected.

7. PLOS authors have the option to publish the peer review history of their article (what does this mean?). If published, this will include your full peer review and any attached files.

Reviewer #2: No

Reviewer #3: No

---

## [Editor Report · Acceptance letter]

9 Oct 2019

PONE-D-19-18659R1 

Can non-pharmacological interventions reduce hospital admissions in people with dementia? A systematic review 

Dear Dr. Packer:

I am pleased to inform you that your manuscript has been deemed suitable for publication in PLOS ONE. Congratulations! Your manuscript is now with our production department. 

With kind regards,

on behalf of

Dr. Wisit Cheungpasitporn 

Academic Editor

PLOS ONE